# Real-Time TDM-Based Expert Clinical Pharmacological Advice Program for Attaining Aggressive Pharmacokinetic/Pharmacodynamic Target of Continuous Infusion Meropenem in the Treatment of Critically Ill Patients with Documented Gram-Negative Infections Undergoing Continuous Veno-Venous Hemodiafiltration

**DOI:** 10.3390/antibiotics12101524

**Published:** 2023-10-10

**Authors:** Milo Gatti, Matteo Rinaldi, Tommaso Tonetti, Antonio Siniscalchi, Pierluigi Viale, Federico Pea

**Affiliations:** 1Department of Medical and Surgical Sciences, Alma Mater Studiorum University of Bologna, 40138 Bologna, Italy; mat.rinaldi1989@gmail.com (M.R.); tommaso.tonetti@unibo.it (T.T.); pierluigi.viale@unibo.it (P.V.); federico.pea@unibo.it (F.P.); 2Clinical Pharmacology Unit, Department for Integrated Infectious Risk Management, IRCCS Azienda Ospedaliero-Universitaria of Bologna, 40138 Bologna, Italy; 3Infectious Disease Unit, Department for Integrated Infectious Risk Management, IRCCS Azienda Ospedaliero-Universitaria of Bologna, 40138 Bologna, Italy; 4Division of Anesthesiology, Department of Anesthesia and Intensive Care, IRCCS Azienda Ospedaliero-Universitaria di Bologna, 40138 Bologna, Italy; 5Anesthesia and Intensive Care Medicine, IRCCS Azienda Ospedaliero-Universitaria di Bologna, 40138 Bologna, Italy; antonio.siniscalchi@aosp.bo.it

**Keywords:** meropenem, continuous infusion, continuous veno-venous hemodiafiltration, continuous renal replacement therapy, Gram-negative infections, PK/PD target attainment, microbiological outcome

## Abstract

(1) Objectives: to describe the pharmacokinetic/pharmacodynamic (PK/PD) profile of continuous infusion (CI) meropenem in critical patients with documented Gram-negative infections undergoing continuous veno-venous hemodiafiltration (CVVHDF) and to assess the relationship with microbiological outcome. (2) Methods: Data were retrospectively retrieved for patients admitted to the general and the post-transplant intensive care units in the period October 2022–May 2023 who underwent CVVHDF during treatment with CI meropenem optimized by means of a real-time therapeutic drug monitoring (TDM)-based expert clinical pharmacological advice (ECPA) program for documented Gram-negative infections. Steady-state meropenem plasma concentrations were measured, and the free fractions (*f*C_ss_) were calculated. Meropenem total clearance (CL_tot_) was calculated at each TDM assessment, and the impact of CVVHDF dose intensity and of residual diuresis on CL_tot_ was investigated by means of linear regression. Optimal meropenem PK/PD target attainment was defined as an *f*C_ss_/MIC ratio > 4. The relationship between meropenem PK/PD target attainment and microbiological outcome was assessed. (3) Results: A total of 24 critical patients (median age 68 years; male 62.5%) with documented Gram-negative infections were included. Median (IQR) meropenem *f*C_ss_ was 19.9 mg/L (17.4–28.0 mg/L). Median (IQR) CL_tot_ was 3.89 L/h (3.28–5.29 L/h), and median (IQR) CVVHDF dose intensity was 37.4 mL/kg/h (33.8–44.6 mL/kg/h). Meropenem dosing adjustments were provided in 20 out of 24 first TDM assessments (83.3%, all decreases) and overall in 26 out of the 51 total ECPA cases (51.0%). Meropenem PK/PD target attainment was always optimal, and microbiological eradication was achieved in 90.5% of assessable cases. (4) Conclusion: the real-time TDM-guided ECPA program was useful in attaining aggressive PK/PD targeting with CI meropenem in critically ill patients undergoing high-intensity CVVHDF and allowed microbiological eradication in most cases with dosing regimens ranging between 125 and 500 mg q6h over 6 h.

## 1. Introduction

Sepsis and septic shock are major life-threatening infection-related conditions affecting critically ill patients, which may often cause multiorgan failure and may account for high mortality rates and massive antibiotic consumption [1]. 

The *Enterobacterales* and some non-fermenting pathogens like *P. aeruginosa* and *A. baumannii* are the Gram-negatives responsible for the majority of the cases of sepsis/septic shock reported among critically ill patients admitted to the intensive care unit (ICU) [2,3,4]. Unfortunately, nowadays the presence of carbapenem resistance (CR) is rapidly growing among Gram-negatives in the ICU setting [5]. Consequently, meropenem is being progressively superseded by novel beta-lactam/beta-lactamase inhibitors showing valuable activity against CR-Gram-negative infections [6]. According to recent guidelines and guidance, meropenem still remains the first-line option for treating infections caused by extended-spectrum beta-lactamase (ESBL)- and/or by AmpC-producing *Enterobacterales*, and is a valuable option for treating those caused by carbapenem-susceptible *P. aeruginosa* or *A. baumannii* [7,8,9]. Referring to antimicrobial resistance in the EU/EEA (EARS-Net), according to the last ECDC annual epidemiological report, ESBL- and/or AmpC-producing *Enterobacterales* and *P. aeruginosa* strains resistant to ceftazidime and/or to piperacillin–tazobactam accounted for 23.8–55.3% and 19.1–23.4% of isolates, respectively [10]. 

Notably, it was shown that attaining an aggressive pharmacokinetic/pharmacodynamic (PK/PD) target of 100%*f*T_>4–8×MIC_ with continuous infusion (CI) beta-lactams, including meropenem, was associated in critically ill ICU patients with both maximization of clinical efficacy and suppression of resistance development [11,12,13]. Unfortunately, achieving this goal in septic patients may be hampered by the need for continuous renal replacement therapy (CRRT). CRRT is needed in up to 70% of cases of sepsis-related acute kidney injury (AKI) [14,15], and may consistently increase meropenem clearance [16]. In critically ill patients undergoing CRRT, selecting proper meropenem dosing regimens for attaining aggressive PK/PD targets may be hindered by the CRRT operative conditions (i.e., modality for solute removal, type of filter material, effluent flow rate), by the patient’s specific pathophysiological conditions (i.e., residual renal function and/or capillary leak syndrome), and by poor susceptibility of the bacterial pathogen to meropenem (i.e., high minimum inhibitory concentrations [MIC]) [16].

Unfortunately, guidance for choosing proper antimicrobial dosing regimens focused on aggressive PK/PD targets during CRRT is currently lacking [17,18]. In this challenging scenario, implementing a real-time therapeutic drug monitoring (TDM)-based expert clinical pharmacological advice (ECPA) program may be helpful in personalizing the CI meropenem dosing regimen for each critically ill patient undergoing CRRT, according to the so-called antimicrobial therapy puzzle concept [19].

The aim of this study was to assess retrospectively the usefulness of a real-time TDM-based ECPA program for attaining aggressive PK/PD targeting by CI meropenem in the treatment of critically ill patients with documented Gram-negative infections undergoing continuous veno-venous hemodiafiltration (CVVHDF).

## 2. Results

Overall, 24 critically ill patients had CI meropenem dosing regimens personalized by means of the TDM-based ECPA for treating documented Gram-negative infections while undergoing CVVHDF. Demographics and clinical features of the patients are summarized in Table 1. Case-by-case features of the included patients are described in Table 2. 

Median [interquartile range (IQR)] age was 68 years (61–74 years), with a male preponderance (62.5%). Median (IQR) Charlson Comorbidity Index (CCI) score was 5 (3–6). Bowel perforation (7 cases; 29.2%) and orthotopic liver transplant (6 cases; 25.0%) were the most frequent underlying diseases. Median (IQR) sequential organ failure assessment (SOFA) score at infection onset was 14 (10.75–17). In total, 21 out of 24 patients (87.5%) underwent invasive mechanical ventilation, and cardiovascular support with vasopressors was required in 83.3% of cases.

CVVHDF was always performed by means of a Prisma flex system equipped with an AN69 high-flux ST-150 filter membrane, and regional anticoagulation was achieved by means of citrate. Median (IQR) blood flow rate (Q_b_) and total effluent flow rate were 150 mL/min (150–150 mL/min) and 2800 mL/h (2615–2900 mL/h), respectively. Median (IQR) CVVHDF dose was 37.4 mL/kg/h (33.8–44.6 mL/kg/h). Replacement solution was always delivered post-filter. In total, 18 out of 24 patients (75.0%) had residual 24-h urinary output with a median (IQR) of 140 mL (40.0–417.5 mL), and values higher than 500 mL were reported in three cases (12.5%).

Infection types were ventilator-associated pneumonia (VAP) in 9/24 cases (37.5%), bloodstream infection (BSI) in 7/24 cases (29.2%), complicated intrabdominal infection (cIAI) plus BSI in 3/24 cases (12.5%), cIAI and cIAI plus VAP in 2/24 cases each (8.3%), and VAP plus BSI in one case (4.2%). Overall, 27 Gram-negative pathogens were isolated, with *Escherichia coli* (25.9%), *Klebsiella pneumoniae* (25.9%), *Enterobacter cloacae* (18.5%), and *Proteus mirabilis* (11.2%) being the most frequent. ESBL and/or AmpC producers accounted for 17 out of 24 (70.8%) *Enterobacterales* clinical isolates. MIC values of meropenem ranged from 0.12 mg/L to 1 mg/L. 

Median (IQR) daily CI maintenance dose (MD) of meropenem was 500 q8h over 8 h (250–1000 q6h over 6 h). Combination therapy was implemented in 10 out of 24 patients (41.7%), with tigecycline and with ciprofloxacin in 9 and in 1 case, respectively. Median (IQR) treatment duration was 10.5 days (6–17 days). Median (IQR) free steady-state concentrations (*f*C_ss_) of meropenem were 19.9 mg/L (17.4–28.0 mg/L), and meropenem total CL (CL_tot_) was 3.89 L/h (3.28–5.29 L/h). Meropenem CL_tot_ correlated poorly both with CVVHDF dose intensity (*r* = 0.22; *p* = 0.13) and with residual diuresis (*r* = 0.02; *p* = 0.87).

A total of 51 TDM-based ECPA recommendations for meropenem were carried out, with a median (IQR) of 1 (1–3) per patient. Dosing adjustments were recommended in 20 out of 24 first TDM assessments (83.3%, all decreases), and overall in 26 out of 51 total ECPA cases (51.0%, of which 2.0% increases and 49.0% decreases). Meropenem PK/PD target attainment was optimal in all of the cases (100.0%) with a median (IQR) *f*C_ss_/MIC ratio of 147.2 (48.5–196.9). 

Microbiological outcome was assessable in 21/24 cases. Microbiological eradication was achieved in 19/21 cases (90.5%). Microbiological failure occurred in two cases (9.5%), namely in one with bacteraemic cIAI due to *Enterobacter cloacae*, and in another with VAP caused by *Enterobacter cloacae*. No cases of resistance developed. Clinical cure was achieved in 54.2% of patients, and the 30-day mortality rate was 50.0%.

The TDM-based meropenem dosing ECPA provided in our study is summarized in Table 3.

## 3. Discussion

Our study described the attainment of aggressive PK/PD targeting by CI meropenem by means of a real-time TDM-based ECPA program in a cohort of critically ill patients with documented severe Gram-negative infections undergoing high-intensity CVVHDF.

Several studies have assessed the population PK of meropenem in critical patients undergoing CRRT, and were recently meta-analyzed [20]. Interestingly, median meropenem CL_tot_ was approximately 25% higher in our study than in previous studies (3.89 L/h vs. 3.03 L/h) [20]. This finding could be explained by the high dose intensity of CVVHDF applied, which was approximately two-thirds higher than reported in previous studies (median 37.4 vs. 25.0 mL/kg/h). Meropenem CL_tot_ was far lower in our cohort compared with healthy subjects (15.54 L/h) [21]. This is in agreement with the severity of the sepsis-related AKI and the multi-organ failure affecting these patients, as witnessed by the relevant proportion of cases having anuria and high SOFA scores at infection onset. 

The lack of correlation between CVVHDF dose intensity and meropenem CL_tot_ is not unexpected, being consistent with what has been previously reported [22,23,24,25,26]. In our study, this could be explained on the one hand by wide intra- and interindividual differences regarding filter age and/or drug adsorption [22], and on the other hand by the fact that the almost homogeneous dose intensity of CVVHDF applied in all of the patients precluded the possibility of showing consistent differences of meropenem CL_tot_ among cases.

The lack of significant correlation between meropenem CL_tot_ and residual diuresis is inconsistent with previous studies [22,27]. This might be explained by the fact that only approximately 10% of our patients had residual diuresis, but we should also recognize that measuring 24 h urinary creatinine clearance rather than residual diuresis might have enabled more reliable assessment of residual renal function [28]. 

The most interesting finding of our study was that CI meropenem doses ranging from 125 to 500 mg q6h over 6 h attained aggressive PK/PD targets in all of the cases undergoing high-intensity CVVHDF and allowed microbiological eradication in the vast majority of these. CI may be the best administration mode for attaining aggressive PK/PD targets with beta-lactams under the same daily dose [29,30]. In this scenario, adopting a real-time TDM-guided ECPA program could be very valuable in allowing prompt dose reduction whenever clinical isolates have wild-type MICs, as occurred in more than 80% of first TDM assessments in the current study, thus also minimizing potential adverse events associated with drug overexposure [31,32].

This may further support the role that a “patient-centered” approach may have in selecting antimicrobial dosing regimens for critically ill patients undergoing CRRT, by taking into account all of the potential variables hindering the aggressive PK/PD target attainment of a specific agent [16]. 

We believe that our findings could also provide some useful suggestions for attaining aggressive PK/PD targeting with CI meropenem in patients with documented Gram-negative infections susceptible to meropenem requiring CVVHDF support with dose intensity similar to ours, for intensive care physicians in centers where real-time TDM is unfeasible. Specifically, the suggested dosing regimens could range from 125 mg q6h over 6 h in cases of clinical isolates with an MIC value ≤ 0.12 mg/L, up to 250 mg q6h over 6 h in cases of clinical isolates with MIC values ranging 0.25–1 mg/L, and 500 mg q6h over 6 h in cases of clinical isolates with MIC values at the clinical breakpoint of 2 mg/L.

We recognize that our study has some limits. The retrospective monocentric study design and the limited sample size should be acknowledged. Only total meropenem concentrations were measured, and the free fractions were estimated by means of the plasma protein binding reported in healthy subjects; with these values being low, the bias should be only marginal. Residual renal function was not assessed by means of 24 h measured urinary creatinine clearance. The role of combo-therapy on microbiological outcome could not be ruled out. Finally, our findings may not be reliable enough in cases of low-intensity CVVHDF or other CRRT modalities.

## 4. Materials and Methods

### 4.1. Study Design

This retrospective study investigated critically ill patients who, in the period between 1 October 2022 and 31 May 2023, were admitted to the general ICU or the post-transplant ICU of the IRCCS Azienda Ospedaliero-Universitaria of Bologna, Italy, and were treated with CI meropenem optimized by means of real-time TDM-guided ECPA for documented Gram-negative infections while undergoing CVVHDF. Patients affected by COVID-19 were excluded.

### 4.2. Data Collection

Demographic (age, sex weight, height, body mass index [BMI]) and clinical/laboratory data (CCI, underlying disease, SOFA score at infection onset, need for mechanical ventilation and/or for vasopressor support) were retrieved for each patient. Clinical isolates, MIC values for meropenem, type/site of infection, treatment duration, meropenem dosage and average plasma C_ss_, eventual co-administration of other anti-Gram-negative active antimicrobial agents, overall number of ECPAs, ECPA-recommended dosing adjustments at first and subsequent TDM assessment, microbiological and clinical outcomes, and 30-day mortality rate were also collected. 

CVVHDF operative conditions were retrieved at each TDM assessment. Specifically, data concerning type of filter, selected anticoagulant, Q_b_, pre-blood pump (PBP) fluid rate, dialysate flow rate (Q_d_), percentage of pre-/post-dilution, replacement fluid rate, and net removal rate per hour were collected. Furthermore, data on 24 h residual diuresis on the day of TDM assessment were also extracted. The total effluent flow rate was defined according to the following equation: pre-filter replacement fluid rate + post-filter replacement fluid rate + net removal rate + PBP fluid rate + Q_d_, as previously reported [33]. CVVHDF dose intensity was calculated as the total effluent flow rate normalized to body weight. 

Site of infection was defined according to the Centers for Disease Control and Prevention (CDC) criteria [34]. Documented BSI was defined as the isolation of a Gram-negative pathogen from at least one blood culture [34]. Documented VAP was defined as the isolation of one or more Gram-negative pathogens with a bacterial load ≥10^4^ CFU/mL in the bronchoalveolar lavage (BAL) fluid culture after > 48 h from endotracheal intubation and initiation of mechanical ventilation in patients showing new or progressive lung infiltrates [35,36]. cIAI was defined as the isolation of one or more Gram-negative pathogens from the peritoneal fluid in patients with infection extended to multiple organs into the peritoneal space [36,37].

The MIC values for meropenem against Gram-negative clinical isolates (*Enterobacterales* and/or *Pseudomonas aeruginosa*) were measured by means of a semi-automated broth microdilution method (Microscan Beckman NMDRM1) and interpreted according to the European Committee on Antimicrobial Susceptibility Testing (EUCAST) clinical breakpoints [38]. Pathogens were considered as meropenem-susceptible whenever the MIC value was ≤2 mg/L.

### 4.3. Meropenem Administration and Sampling Procedure

Meropenem was prescribed by the treating intensive care physicians and/or the infectious disease consultant as first- or second-line therapy according to results of antimicrobial susceptibility tests. Therapy was always started with a loading dose (LD) of 2 g over 2 h infusion, followed by an initial MD administered by CI. Aqueous solutions were reconstituted every 6–8 h and infused over 6–8 h in order to prevent meropenem degradation, as recommended [39,40]. 

The initial MD regimen was defined on a case-by-case basis in relation to CVVHDF conditions, presence/absence of residual diuresis, site of infection, and underlying clinical conditions. Dosing was subsequently optimized by means of a real-time TDM-guided ECPA program. For this purpose, blood samples for measuring meropenem C_ss_ were first collected after at least 24 h from starting therapy, and then reassessed every 48–72 h whenever feasible. Total meropenem plasma concentrations were determined by means of a validated liquid chromatography–tandem mass spectrometry method [12]. Precision and accuracy were assessed by replicate analysis of quality control samples against calibration standards. The intra- and inter-assay coefficients of variations were <±15%, as required by the EMA guidelines. The lower limit of quantification was 0.3 mg/L. The assay was linear in the range from 0.3 to 84.4 mg/L. At each TDM assessment, meropenem CL_tot_ was calculated by means of the following formula: CL_tot_ (L/h) = infusion rate (mg/h)/C_ss_ (mg/L).

Real-time expert interpretation of meropenem TDM results was performed by well-trained MD clinical pharmacologists (ECPA) with long-standing expertise who suggested dosing adaptation whenever needed. The TDM-guided ECPA was structured as previously reported [19].

### 4.4. Relationship between Meropenem Aggressive PK/PD Target and Microbiological Outcome

The percentage of time of CI meropenem *f*C_ss_ above the MIC was selected as the best PK/PD parameter for efficacy and defined as the *f*C_ss_/MIC ratio (equivalent to % *f*T_>MIC_). Considering that only total meropenem concentrations were measured, the *f* fraction was calculated by multiplying total meropenem C_ss_ by 0.98, according to a 2% plasma protein binding [41]. 

Aggressive PK/PD target attainment was defined as optimal when the *f*C_ss_/MIC ratio was >4 (equivalent to 100%*f*T_>4×MIC_), and quasi-optimal or suboptimal when *f*C_ss_/MIC ratio was 1–4 or <1 (equivalent to 100%*f*T_1–4×MIC_ and to <100%*f*T_1×MIC_), respectively, as previously reported [42]. Specifically, a *f*C_ss_/MIC > 4 is equal to *f*C_ss_ > 4 × MIC. Considering that meropenem was administered by CI in all included patients, it may be supposed that C_ss_ were stable and constant throughout the 24 h. Consequently, the *f*C_ss_ > 4 × MIC may be expressed as 100%*f*T_>4×MIC_, as previously reported [12].

In patients undergoing more than one TDM-guided ECPA recommendation, average meropenem *f*C_ss_ was considered by calculating the mean of all of the observed C_ss_ values (the first one before any dosage adjustment and the subsequent ones after eventual dosage adjustments).

This threshold was selected according to in vitro, experimental animal, and clinical studies showing that attaining aggressive PK/PD targets of C_ss_/MIC ratios ≥ 4 (equivalent to 100%*f*_>4×MIC_) with beta-lactams may be associated both with maximization of clinical efficacy and with microbiologic eradication and prevention of resistance development against Gram-negative pathogens [11,12,13,43].

Optimization of meropenem dosing was performed as previously reported [19]. Specifically, a 25% or 50% dosing decrease was adopted whenever the C_ss_/MIC ratio was equal to 8–10 or above 10, respectively; dosing was confirmed whenever the C_ss_/MIC ratio was equal to 4–8; and 25% or 50% dosing increase was implemented whenever the C_ss_/MIC ratio was equal to 2–4 or below 2, respectively.

Microbiological eradication and clinical outcome were assessed for each patient and correlated with meropenem PK/PD target attainment. Microbiological eradication or failure was defined, respectively, as the absence or the persistence of the same Gram-negative pathogen isolated in the index cultures in at least one subsequent assessment after starting meropenem therapy. Resistance development was defined as an increase in the meropenem MIC value against the original clinical isolate beyond the EUCAST clinical breakpoint of susceptibility. Primary outcome was microbiological eradication, whereas clinical cure (defined as the complete resolution of signs and symptoms of the infection coupled with documented microbiological eradication at the end of treatment and absence of recurrence or relapse at 30-day follow-up and/or resistance development) and 30-day mortality rate represented the secondary outcomes. 

### 4.5. Statistical Analysis

Continuous data are expressed as median and interquartile range (IQR), whereas categorical variables are presented as counts or percentages. The relationships between CVVHDF dose intensity or residual diuresis and meropenem CL_tot_ were assessed by linear regression, and the Pearson’s r value was calculated. *p* values < 0.05 were considered statistically significant. Statistical analysis was performed by using MedCalc for Windows (MedCalc statistical software, version 19.6.1, MedCalc Software Ltd., Ostend, Belgium).

## 5. Conclusions

In conclusion, our findings suggested that a real-time TDM-guided ECPA program may be useful in attaining aggressive PK/PD targeting with CI meropenem in critically ill patients undergoing high-intensity CVVHDF and that dosing regimens ranging between 125 and 500 mg q6h over 6 h may allow microbiological eradication in most cases. The profile of the susceptibility test results of the Gram-negative isolates to meropenem may be a major factor affecting the magnitude of the CI meropenem dosing regimen needed when very homogeneous operative conditions for CVVHDF are applied. Large prospective studies are warranted to confirm our findings.

## Figures and Tables

**Table 1 antibiotics-12-01524-t001:** Demographics and clinical characteristics of included patients.

Demographics and Clinical Variables	Patients (N = 24)
Patient demographics	
Age (years) [median (IQR)]	68 (61–74)
Gender (male/female) [n (%)]	15/9 (62.5/37.5)
Caucasian [n (%)]	23 (95.8)
Body weight (Kg) [median (IQR)]	67.5 (60.0–80.0)
Body mass index (Kg/m^2^) [median (IQR)]	24.0 (22.0–27.6)
Charlson Comorbidity Index [median (IQR)]	5 (3–6)
Underlying diseases [n (%)]	
Bowel perforation	7 (29.2)
OLTx	6 (25.0)
Acute-on-chronic liver failure	3 (12.5)
Cholangitis/cholecystitis	3 (12.5)
Acute pulmonary edema	2 (8.3)
Others *	3 (12.5)
Severity of infections	
SOFA score at infection onset [median (IQR)]	14 (10.75–17)
Mechanical ventilation [n (%)]	21 (87.5)
Vasopressors [n (%)]	20 (83.3)
CVVHDF settings	
Total effluent flow rate (mL/h) [median (IQR)]	2800 (2615–2900)
CVVHDF dose intensity (mL/kg/h) [median (IQR)]	37.4 (33.8–44.6)
Blood flow rate (mL/min) [median (IQR)]	150 (150–150)
Net removal (mL/h) [median (IQR)]	90 (50–120)
Residual diuresis (mL/24 h) [median (IQR)]	140 (40.0–417.5)
Site of infection [n (%)]	
VAP	9 (37.5)
BSI	7 (29.2)
cIAI + BSI	3 (12.5)
cIAI	2 (8.3)
cIAI + VAP	2 (8.3)
VAP + BSI	1 (4.2)
Gram-negative clinical isolates ^1^ [n (%)]	
*Escherichia coli*	7 (25.9)
*Klebsiella pneumoniae*	7 (25.9)
*Enterobacter cloacae*	5 (18.5)
*Proteus mirabilis*	3 (11.2)
*Pseudomonas aeruginosa*	2 (7.4)
*Klebsiella oxytoca*	1 (3.7)
*Klebsiella aerogenes*	1 (3.7)
*Acinetobacter baumannii*	1 (3.7)
Antibiotic treatment	
Meropenem daily dose (mg) [median (IQR)]	500 q8h over 8 h (250–1000 q6h over 6 h)
Meropenem *f*C_ss_ (mg/L) [median (IQR)]	19.9 (17.4–28.0)
Meropenem CL_tot_ (L/h) [median (IQR)]	3.89 (3.28–5.29)
Combination therapy [n (%)] Tigecycline Ciprofloxacin	10 (41.7) 9 1
Treatment duration (days) [median (IQR)]	10.5 (6–17)
PK/PD target attainment	
Meropenem *f*C_ss_/MIC [median (IQR)]	147.2 (48.5–196.9)
*f*C_ss_/MIC > 4 [n (%)] *f*C_ss_/MIC = 1–4 [n (%)] *f*C_ss_/MIC < 1 [n (%)]	24 (100.0) 0 (0.0) 0 (0.0)
TDM-based ECPA	
Overall TDM-based ECPA	51
N. of TDM-based ECPA per patient [median (IQR)]	1 (1–3)
N. of dosage confirmations at first TDM assessment [n (%)]	4 (16.7)
N. of dosage increases at first TDM assessment [n (%)]	0 (0.0)
N. of dosage decreases at first TDM assessment [n (%)]	20 (83.3)
Overall n. of dosage confirmations [n (%)]	25 (49.0)
Overall n. of dosage increases [n (%)]	1 (2.0)
Overall n. of dosage decreases [n (%)]	25 (49.0)
Clinical outcome	
Microbiological eradication ^2^ [n (%)]	19 (90.5)
Resistance development ^2^ [n (%)]	0 (0.0)
Clinical cure [n (%)]	13 (54.2)
30-day mortality rate [n (%)]	12 (50.0)

* including obstructive acute kidney injury (n = 1), DRESS syndrome (n = 1), and febrile neutropenia (n = 1). ^1^ A total of 27 pathogens were isolated. ^2^ Assessable in 21 out of 24 cases. BSI: bloodstream infection; cIAI: complicated intrabdominal infection; CL_tot_: total clearance; C_ss_: steady-state concentration; CVVHDF: continuous veno-venous hemodiafiltration; ECPA: expert clinical pharmacological advice; IQR: interquartile range; MIC: minimum inhibitory concentration; OLTx: orthotopic liver transplant; PK/PD: pharmacokinetic/pharmacodynamic; SOFA: sequential organ failure assessment; TDM: therapeutic drug monitoring; VAP: ventilator-associated pneumonia.

**Table 2 antibiotics-12-01524-t002:** Case-by-case demographic and clinical features of 24 critically ill patients with documented Gram-negative infections treated with CI meropenem according to a TDM-guided ECPA program during CVVHDF support.

ID Case	Age/Gender	Underlying Disease	CCI	MV/ Vasopressors	Baseline SOFA Score	Type of Infection	Pathogen	MIC (mg/L)	Initial Meropenem MD	Average *f*Css (mg/L)	Average *f*Css/MIC Ratio	PK/PD Target Attainment	Combination Therapy	Microbiological Eradication	Clinical Cure	30-Day Mortality Rate
#1	54/M	ACLF	5	Yes/Yes	19	BSI	ESBL-producing *Escherichia coli*	0.12	500 mg q6h over 6 h	19.11	159.25	Optimal	None	Yes	No ***	Yes
#2	77/F	Cholangitis	3	Yes/Yes	15	cIAI + BSI	*Enterobacter cloacae*	0.12	1000 mg q6h over 6 h	22.91	190.94	Optimal	Tigecycline	No	No	No
#3	67/F	OLTx	5	Yes/Yes	16	VAP	*Enterobacter cloacae*	1	1000 mg q6h over 6 h	44.59	44.59	Optimal	Tigecycline	No	No	Yes
#4	70/F	Bowel perforation	3	Yes/Yes	17	BSI	ESBL-producing *Escherichia coli*	0.12	1000 mg q6h over 6 h	75.66	630.47	Optimal	None	NA	No	Yes
#5	67/F	OLTx	5	Yes/Yes	16	VAP	*Enterobacter cloacae*	1	1000 mg q6h over 6 h	30.02	30.02	Optimal	None	Yes	No ***	Yes
#6	77/F	Cholangitis	3	Yes/Yes	16	cIAI + BSI	*Enterobacter cloacae*	0.12	250 mg q6h over 6 h	8.43	70.23	Optimal	Tigecycline	Yes	Yes	No
#7	63/M	OLTx	6	Yes/Yes	17	cIAI + BSI	*Klebsiella oxytoca*	0.12	1000 mg q6h over 6 h	18.54	148.31	Optimal	Tigecycline	Yes	Yes	No
#8	62/M	Acute pulmonary oedema	6	Yes/No	6	VAP	ESBL-producing *Klebsiella pneumoniae*	0.5	500 mg q6h over 6 h	20.91	41.81	Optimal	Ciprofloxacin	Yes	Yes	No
#9	63/M	OLTx	6	Yes/Yes	17	VAP	*Acinetobacter baumannii*	1	1000 mg q6h over 6 h	27.59	27.59	Optimal	Tigecycline	Yes	Yes	No
#10	57/F	OLTx	5	Yes/No	12	VAP	*Proteus mirabilis*	0.12	1000 mg q6h over 6 h	18.69	155.78	Optimal	None	Yes	Yes	No
#11	81/M	Acute pulmonary oedema	5	Yes/Yes	11	VAP	ESBL-producing *Klebsiella pneumoniae*	0.12	1000 mg q6h over 6 h	42.24	351.98	Optimal	None	Yes	No ***	Yes
#12	58/M	ACLF	7	No/Yes	15	BSI	*Klebsiella pneumoniae*	0.12	1000 mg q6h over 6 h	30.09	250.72	Optimal	None	Yes	Yes	Yes
#13	62/M	Bowel perforation	2	Yes/Yes	18	cIAI	ESBL-producing *Escherichia coli*	0.12	250 mg q6h over 6 h	5.98	48.92	Optimal	None	Yes	No ***	Yes
#14	41/M	ACLF	3	No/No	8	BSI	ESBL-producing *Escherichia coli* ESBL-producing *Klebsiella pneumoniae*	0.12 0.12	750 mg q6h over 6 h	25.77	214.78	Optimal	None	Yes	Yes	No
#15	73/M	Bowel perforation	4	Yes/Yes	9	cIAI + VAP	ESBL-producing *Proteus mirabilis*	0.12	1000 mg q6h over 6 h	29.20	233.63	Optimal	Tigecycline	Yes	No ***	Yes
#16	74/M	Bowel perforation	6	Yes/Yes	12	cIAI + VAP	*Escherichia coli* *Proteus mirabilis*	0.12	1000 mg q6h over 6 h	27.34	218.74	Optimal	None	NA	No	Yes
#17	71/M	Obstructive AKI	9	Yes/Yes	18	VAP	ESBL-producing *Klebsiella pneumoniae*	0.12	1000 mg q6h over 6 h	8.82	70.56	Optimal	None	NA	No	Yes
#18	71/M	Bowel perforation	5	Yes/Yes	9	BSI	ESBL-producing *Klebsiella pneumoniae*	1	1000 mg q6h over 6 h	16.82	16.82	Optimal	None	Yes	Yes	Yes
#19	69/M	Dress syndorme	9	Yes/Yes	11	VAP	AmpC-producing *Klebsiella aerogenes*	0.12	500 mg q6h over 6 h	14.70	122.50	Optimal	Tygeciccline	Yes	Yes	No
#20	83/F	Bowel perforation	7	Yes/Yes	20	VAP + BSI	ESBL-producing *Escherichia coli*	0.12	1000 mg q6h over 6 h	17.54	146.18	Optimal	None	Yes	No ***	Yes
#21	50/F	OLT	4	Yes/Yes	12	BSI	*Pseudomonas aeruginosa*	0.25	250 mg q6h CI	20.04	80.16	Optimal	Tygeciccline	Yes	Yes	No
#22	77/M	Bowel perforation	7	Yes/No	9	cIAI	*Escherichia coli* *Enterobacter cloacae*	0.12 0.12	1000 mg q6h over 6 h	17.95	149.61	Optimal	Tygeciccline	Yes	Yes	No
#23	56/M	Febrile neutropenia	3	No/Yes	13	BSI	ESBL-producing *Escherichia coli*	0.12	1000 mg q6h over 6 h	19.83	158.63	Optimal	None	Yes	Yes	No
#24	74/F	Cholecistitis	3	Yes/Yes	10	VAP	*Pseudomonas aeruginosa*	1	500 mg q6h over 6 h	14.80	14.80	Optimal	None	Yes	Yes	No

* case #1: ESBL-producing *E. coli* was eradicated from follow-up blood culture, but signs and symptoms of the infection persisted, resulting in clinical failure and subsequent implementation of compassionate care; case #5: *Enterobacter cloacae* was eradicated from follow-up BAL, but signs and symptoms of the infection persisted, resulting in clinical failure and mortality; case #11: ESBL-producing *K.pneumoniae* was eradicated from follow-up BAL, but signs and symptoms of infection persisted, resulting in clinical failure and subsequent implementation of compassionate care; case #13: ESBL-producing *E. coli* was eradicated from peritoneal fluid, but fever and need for vasopressors persisted; case #15: ESBL-producing *P. mirabilis* was eradicated from peritoneal fluid, but fever and need for vasopressors persisted, resulting in implementation of compassionate care due to inoperability; case #20: ESBL-producing *E. coli* was eradicated from peritoneal fluid, but fever and need for vasopressors persisted, resulting in implementation of compassionate care due to underlying conditions. ACLF: acute-on-chronic liver failure; AKI: acute kidney injury; BSI: bloodstream infection; CCI: Charlson Comorbidity Index; cIAI: intrabdominal infection; Css: steady-state concentrations; ECPA: expert clinical pharmacological advice; ESBL: extended-spectrum beta-lactamase; MD, maintenance dose; MIC: minimum inhibitory concentration; MV: mechanical ventilation; NA: not assessed; OLTx: orthotopic liver transplant; PK/PD: pharmacokinetic/pharmacodynamic; SOFA: sequential organ failure assessment; TDM: therapeutic drug monitoring; VAP: ventilator-associated pneumonia. PK/PD target column: green box: optimal PK/PD target; yellow box: quasi-optimal PK/PD target; red box: suboptimal PK/PD target. Microbiological eradication column: green box: microbiological eradication; red box: microbiological failure; grey box: not assessable.

**Table 3 antibiotics-12-01524-t003:** Meropenem dosing ECPA provided in our study.

Optimal PK/PD Target	MIC of Isolated Pathogen (mg/L)	TDM-Based Meropenem Dosing ECPA Recommendation
*f*C_ss_/MIC > 4	0.12	125 mg q6h over 6 h
*f*C_ss_/MIC > 4	0.25–1	250 mg q6h over 6 h
*f*C_ss_/MIC > 4	2	500 mg q6h over 6 h

CI: continuous infusion; C_ss_: steady-state concentration; MIC: minimum inhibitory concentration; PK/PD: pharmacokinetic/pharmacodynamic

## Data Availability

The data presented in this study are available on request from the corresponding author. The data are not publicly available due to privacy concerns.

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
