# Peer review of "Real-Time TDM-Based Expert Clinical Pharmacological Advice Program for Attaining Aggressive Pharmacokinetic/Pharmacodynamic Target of Continuous Infusion Meropenem in the Treatment of Critically Ill Patients with Documented Gram-Negative Infections Undergoing Continuous Veno-Venous Hemodiafiltration"

_antibiotics, 2023, doi:10.3390/antibiotics12101524_

Round 1
Reviewer 1 Report
Authors investigated the pharmacokinetics/pharmacodynamics profile of continous infusion meropenem in critical patients with documented Gram-negative infections requring continuous veno-venous hemodiafiltration, and assess the relateionship with microbiological and clinical outcome. This manuscript is well written, and the results contain important findings in antibiotic therapy. Since there is nothing special to point out regarding the structure, method, results, or discussion of the paper, I judge that this paper is acceptable.
Author Response
Reviewer #1
Q1. Authors investigated the pharmacokinetics/pharmacodynamics profile of continuous infusion meropenem in critical patients with documented Gram-negative infections requiring continuous veno-venous hemodiafiltration, and assess the relationship with microbiological and clinical outcome. This manuscript is well written, and the results contain important findings in antibiotic therapy. Since there is nothing special to point out regarding the structure, method, results, or discussion of the paper, I judge that this paper is acceptable.
A1. We thank the reviewer for appreciating our manuscript.
Reviewer 2 Report
The authors should consider the followings:
Liver function test results should be supplemented, the authors should assess the liver toxicity, with and without the real-time TDM-based expert clinical pharmacological advice program, to evaluate the efficacy of the program.
Risk of bias and limitations of the current study should be listed.
As per expert clinical pharmacological advice (ECPA), the authors may explain how the program select the experts. 1. What criteria were used in the expert selection? 2. Did the experts serving the same program followed a consistent practice guideline? 3. Which practice guideline were referred? 4. Did the authors check the intra and inter expert variability? What were the results if the variability?
The authors may provide the data of the hospital acquired infection of that hospital, of the reporting year(s). Otherwise, the authors supplement that with the regional and/or national results of the the rate of the hospital acquired infection.
Since the data were drawn during the covid pandemic, did the authors factor in covid as the variable(s) and/or other seasonal flu (or infections) in their model of analyses? In addition, did the hospital services or diagnostic and/or treament departments, affect by covid during the said reporting period? Variants and % prevalence of the variants of covid in the reporting period of the region should also be provided, if available.
The authors may provide the information, whether the time and involvement of, expert clinical pharmacological advice (ECPA), being affected by covid, of the reporting period.
Did the experts given adequate time to practice the guideline from program of the expert clinical pharmacological advice (ECPA)? Given the publications from ECPA seems to be around 2022?
In demographics, the authors should list the % Caucasian.
The authors may provide data whether the study show any significant changes of 30-day mortality rate, with and without the real-time TDM-based expert clinical pharmacological advice program.
The authors should clarify and provide eGFR data, and to assess whether the TDM-based expert advice program would be resulted in reducing the renal toxicity, in preserving renal function and/or vice versa.
The authors should provide a list of performance indicators of the expert programme, to monitor and evaluate the said programme and for the development of such programmes.
"Total meropenem plasma concentrations were determined by means of a validated
liquid chromatography-tandem mass spectrometry method." The authors may provide a brief list of validated parameters per the validated LCMS method, i.e. the specificity, sensitivity, LLOQ, dilution reliability, etc, and what standards of validation, did the authors used for LCMS methodology? (Per FDA guideline?)
"f was calculated multiplying total meropenem Css by 0.98 according to a plasma protein binding of 2% reported in the literature" did the author or the referred literature consider the population specific factors, i.e. of aged population, of Caucasian settings?
The authors may provide data for 45-day and/or 90-day, instead of just 30-day mortality rate. Also, the authors should provide definition of the terms, "clinical cure". The authors should give more details for the case, where with yes for "Microbiological eradication", while no for "clinical cure".
The authors required professional english editing.
Author Response
Reviewer #2
The authors should consider the followings:
Q1. Liver function test results should be supplemented, the authors should assess the liver toxicity, with and without the real-time TDM-based expert clinical pharmacological advice program, to evaluate the efficacy of the program.
A1. We thank the reviewer for this comment, allowing us to better clarify the aim of our study. We aimed to assess the usefulness of a real-time TDM-based ECPA program for attaining aggressive PK/PD target of CI meropenem in the specific scenario of critically ill patients with documented Gram-negative infections undergoing continuous veno-venous hemodiafiltration. Notably, at our institution all patients (both critical and non-critical) undergo antimicrobial optimization according to a real-time TDM-based ECPA program, as reported in our previous articles (refer to doi: 10.1016/j.ijantimicag.2023.106884; doi: 10.1186/s13054-022-04050-9; doi: 10.3389/fphar.2021.755075). Consequently, pre-post comparison was unfeasible. We respectfully disagree on the fact that liver function test would be valuable since meropenem is not considered a hepatotoxic drug (refer to DOI: 10.1002/hep.28323).
Q2. Risk of bias and limitations of the current study should be listed.
A2. Thank you for this suggestion. Limitations of our study (i.e., retrospective design, limited sample size, measurement of only total meropenem concentrations, lack of assessment of residual renal function, and the impact of combination therapy on microbiological outcome) were already recognized in the Discussion section (refer to Line 214-221).
Q3. As per expert clinical pharmacological advice (ECPA), the authors may explain how the program select the experts. 1. What criteria were used in the expert selection? 2. Did the experts serving the same program followed a consistent practice guideline? 3. Which practice guideline were referred? 4. Did the authors check the intra and inter expert variability? What were the results if the variability?
A3. We thank the reviewer for this comment allowing us to better clarify this issue. TDM-guided ECPAs were performed by well-trained MD Clinical Pharmacologists with long-standing expertise in the personalization of antimicrobial dosing. In our study, meropenem dosing were adjusted according to recommendations just defined in previously published articles (refer to doi: 10.1016/j.ijantimicag.2023.106884; doi: 10.1186/s13054-022-04050-9; doi: 10.1007/s00134-020-06050-1; doi: 10.1186/s13054-019-2378-9). The approach was always applied by the MD Clinical Pharmacologists involved in the program, so that no intra- and inter-expert variability was expected. Specifically, the adjustments were performed according to the following scheme (refer to doi: 10.1016/j.ijantimicag.2023.106884; doi: 10.1186/s13054-022-04050-9; doi: 10.1007/s00134-020-06050-1; doi: 10.1186/s13054-019-2378-9):
- 50% dosing decrease if Css > 10 x MIC
- 25% dosing decrease if Css = 8-10 x MIC
- dosing confirm if Css = 4-8 x MIC
- 25% dosing increase if Css = 2-4 x MIC
- 50% dosing increase if Css < 2 x MIC
Details of the ECPA program and dosing adjustments were added in the Methods section (refer to Line 313-317).
Q4. The authors may provide the data of the hospital acquired infection of that hospital, of the reporting year(s). Otherwise, the authors supplement that with the regional and/or national results of the the rate of the hospital acquired infection.
A4. Thank you for this suggestion. We added the nationwide prevalence of ESBL-producing Enterobacterales and MDR Pseudomonas aeruginosa retrieved from the last ECDC annual report in the Introduction section (refer to Line 60-64).
Q5. Since the data were drawn during the covid pandemic, did the authors factor in covid as the variable(s) and/or other seasonal flu (or infections) in their model of analyses? In addition, did the hospital services or diagnostic and/or treament departments, affect by covid during the said reporting period? Variants and % prevalence of the variants of covid in the reporting period of the region should also be provided, if available.
A5. Thank you for this comment. The study included only patients admitted to the general and to the post-transplant ICUs. COVID patients, who were admitted to a dedicated COVID-ICU, were excluded. We better specified this aspect in the Methods section (refer to Line 228-229).
Q6. The authors may provide the information, whether the time and involvement of, expert clinical pharmacological advice (ECPA), being affected by covid, of the reporting period.
A6. As previously discussed in response to comment No. 6, COVID patients were excluded.
Q7. Did the experts given adequate time to practice the guideline from program of the expert clinical pharmacological advice (ECPA)? Given the publications from ECPA seems to be around 2022?
A7. We thank the reviewer for this comment, allowing us to better clarify expert interpretation of TDM results. As reported previously (refer to doi: 10.1016/j.ijantimicag.2023.106884; doi: 10.1186/s13054-022-04050-9; doi: 10.3389/fphar.2021.755075), personalization of antimicrobial TDM results is based on the “antimicrobial puzzle” concepts and is carried out by MD Clinical Pharmacologists with specific and long-standing expertise in this dedicated field. The manuscripts were published in the last two years because this program has been implanted at our institution only in early 2021, but this approach was adopted since several years by authors well-before the article publication at another institution (refer among others to doi: 10.1111/bcp.12806; doi: 10.1111/bcpt.12249). We specified more in detail this aspect in the Methods section (refer to Line 290-292).
Q8. In demographics, the authors should list the % Caucasian.
A8. Thank you for this suggestion. We added this information in Table 1.
Q9. The authors may provide data whether the study show any significant changes of 30-day mortality rate, with and without the real-time TDM-based expert clinical pharmacological advice program.
A9. Thank you for this comment. As previously reported in response to comment No. 1, at our institution all patients (both critical and non-critical) undergo to antimicrobial optimization according to a real-time TDM-based ECPA program, as reported in previous articles (refer to doi: 10.1016/j.ijantimicag.2023.106884; doi: 10.1186/s13054-022-04050-9; doi: 10.3389/fphar.2021.755075). Consequently, comparison between real-time TDM-based ECPA program and “standard” approach was unfeasible and was out of the aims of our study.
Q10. The authors should clarify and provide eGFR data, and to assess whether the TDM-based expert advice program would be resulted in reducing the renal toxicity, in preserving renal function and/or vice versa.
A10. Thank you for this comment, allowing us to better clarify this relevant issue. The aim of our study was to assess the usefulness of a real-time TDM-based ECPA program for attaining aggressive PK/PD target of CI meropenem in the specific scenario of critically ill patients with documented Gram-negative infections undergoing continuous renal replacement therapy (specifically continuous veno-venous hemodiafiltration). It should be mentioned that in the scenario of CVVHDF, eGFR is unreliable as marker of residual renal function since renal replacement therapy removes consistently serum creatinine. Notably, as previously discussed in the response to comments No. 1 and 9, at our institution all patients (both critical and non-critical) undergo antimicrobial optimization according to a real-time TDM-based ECPA program, as reported in previous articles (refer to doi: 10.1016/j.ijantimicag.2023.106884; doi: 10.1186/s13054-022-04050-9; doi: 10.3389/fphar.2021.755075). Consequently, comparison between real-time TDM-based ECPA program and “standard” approach was unfeasible and was out of the aims of our study.
Q11. The authors should provide a list of performance indicators of the expert programme, to monitor and evaluate the said programme and for the development of such programmes.
A11. Thank you for this comment, allowing us to better clarify this important issue. Performance indicators (i.e., number of meropenem dosing adjustments at first TDM assessment; number of overall meropenem dosing adjustments) were provided in Table 1 and in the Results section (refer to Line 150-154). For the overall performance indicators of the program in the ICU scenario, please refer to this specific article (doi: 10.1186/s13054-022-04050-9).
Q12. "Total meropenem plasma concentrations were determined by means of a validated liquid chromatography-tandem mass spectrometry method." The authors may provide a brief list of validated parameters per the validated LCMS method, i.e. the specificity, sensitivity, LLOQ, dilution reliability, etc, and what standards of validation, did the authors used for LCMS methodology? (Per FDA guideline?)
A12. Thank you for this comment. The analytic method was validated according to the EMA guidelines, and we added parameter data in the Methods section (refer to Line 284-287).
Q13. "f was calculated multiplying total meropenem Css by 0.98 according to a plasma protein binding of 2% reported in the literature" did the author or the referred literature consider the population specific factors, i.e. of aged population, of Caucasian settings?
A13. Thank you for this comment. As specified in Reference no. 40, 2% protein binding was assessed in a phase I study in healthy male volunteers, and this was recognized as a limit in the Discussion section (refer to Line 215-217).
Q14. The authors may provide data for 45-day and/or 90-day, instead of just 30-day mortality rate. Also, the authors should provide definition of the terms, "clinical cure". The authors should give more details for the case, where with yes for "Microbiological eradication", while no for "clinical cure".
A14. We thank the reviewer for this comment, allowing us to better clarify these different issues. “Clinical cure” was already defined in the Methods section (refer to Line 324-327). We respectfully disagree with the suggestion of providing data on 45-day and/or 90-day mortality since we preferred considering the 28-day/30-day or ICU mortality rate which are the most frequent endpoints of mortality applied in the critical setting, as previously reported (refer to DOI 10.1186/s13054-017-1609-1). As suggested, we added in the footnotes of Table 2 details of cases with documented microbiological eradication and unfavorable clinical outcome.
Q15. The authors required professional english editing.
A15. The Ms. was extensively edited.

Reviewer 3 Report
Your manuscript titled "Real-time TDM-based expert clinical pharmacological advice program for attaining aggressive pharmacokinetic/pharmacodynamic target of continuous infusion meropenem in the treatment of critically ill patients with documented Gram-negative infections undergoing continuous veno-venous hemodiafiltration" is a valuable addition to the literature on critically ill septic patient care. The finding provides strong support to the value of real-time TDM-based ECPA practice. The manuscript is well written and data presented clearly.
Author Response
Reviewer #3
Q1. Your manuscript titled "Real-time TDM-based expert clinical pharmacological advice program for attaining aggressive pharmacokinetic/pharmacodynamic target of continuous infusion meropenem in the treatment of critically ill patients with documented Gram-negative infections undergoing continuous veno-venous hemodiafiltration" is a valuable addition to the literature on critically ill septic patient care. The finding provides strong support to the value of real-time TDM-based ECPA practice. The manuscript is well written and data presented clearly.
A1. We thank the reviewer for appreciating our manuscript.
Reviewer 4 Report
I have concerns regarding the choice of PK/PD target chosen for the study. The authors mention “The percentage of time of meropenem free concentrations above the MIC was selected as the best PK/PD parameter for efficacy and defined as fCss/MIC”.
Meropenem has a time dependent antibacterial efficiency; to obtain an optimal bactericidal activity, the blood concentration must ideally be maintained above MIC during at least 40% of the dosing interval (40% T/MIC).
O'Jeanson A, Larcher R, Le Souder C, Djebli N, Khier S. Population Pharmacokinetics and Pharmacodynamics of Meropenem in Critically Ill Patients: How to Achieve Best Dosage Regimen According to the Clinical Situation. Eur J Drug Metab Pharmacokinet. 2021 Sep;46(5):695-705. doi: 10.1007/s13318-021-00709-w].
Hence, T>MIC is an ideal PK/PD target for meropenem.
While the authors mention the percent of time….as efficacy parameter, but they define it as fCss/MIC, which holds true for antibiotics exhibiting concentration-dependent killing and not meropenem.
Minor editings required.
Author Response
Reviewer #4
Q1. I have concerns regarding the choice of PK/PD target chosen for the study. The authors mention “The percentage of time of meropenem free concentrations above the MIC was selected as the best PK/PD parameter for efficacy and defined as fCss/MIC”.
Meropenem has a time dependent antibacterial efficiency; to obtain an optimal bactericidal activity, the blood concentration must ideally be maintained above MIC during at least 40% of the dosing interval (40% T/MIC).
O'Jeanson A, Larcher R, Le Souder C, Djebli N, Khier S. Population Pharmacokinetics and Pharmacodynamics of Meropenem in Critically Ill Patients: How to Achieve Best Dosage Regimen According to the Clinical Situation. Eur J Drug Metab Pharmacokinet. 2021 Sep;46(5):695-705. doi: 10.1007/s13318-021-00709-w].
Hence, T>MIC is an ideal PK/PD target for meropenem.
While the authors mention the percent of time….as efficacy parameter, but they define it as fCss/MIC, which holds true for antibiotics exhibiting concentration-dependent killing and not meropenem.
A1. We thank the review for this comment, allowing us to better clarify this relevant issue. We fully agree with the reviewer that meropenem has time-dependent activity and that %T>MIC represents the best PK/PD parameter for efficacy. We would like to highlight that fCss/MIC ratio (average steady-state drug concentration over MIC) and %fT>MIC (duration of time that the free drug concentration remains above the MIC) have the same meaning, but simply transposed in different formulas. Indeed, a fCss/MIC ratio of 1 is equivalent to 100%fT>MIC, and a fCss/MIC ratio of 4 is equivalent to 100%fT>4 x MIC. We better clarify this issue in Methods section (refer to Line 296-303).
Round 2
Reviewer 4 Report
I still do not concur with the author's choice of PK/PD target for the study. The authors have given a clarification for the same, request them to provide an appropriate reference for the same after which I would like to re-consider.
Besides this, as stated by authors, I can't find the clarification in the Methods section.
Minor editing required.
Author Response
Q1. I still do not concur with the author's choice of PK/PD target for the study. The authors have given a clarification for the same, request them to provide an appropriate reference for the same after which I would like to re-consider. Besides this, as stated by authors, I can't find the clarification in the Methods section.
A1. We thank the review for this comment. We have tried to better clarify this relevant issue. As previously reported, we would like to highlight that fCss/MIC ratio (average steady-state drug concentration over MIC) and %fT>MIC (duration of time that the free drug concentration remains above the MIC) have the same meaning, but simply transposed in different formulas.
Mathematically, a fCss/MIC > 4 is equal to fCss > 4 x MIC. Considering that meropenem was administered by CI in all included patients, it may be supposed that steady-state concentrations (Css) were stable and constant throughout the 24 hours. Consequently, the fCss > 4 x MIC is equal to 100%fT>4 x MIC when CI is applied. This concept was previously applied in several studies, also including meropenem (refer to doi: 10.3389/fphar.2021.781892; doi: 10.3390/pharmaceutics14081585; doi: 10.3390/pharmaceutics12090785; doi: 10.3390/antibiotics10111311; doi: 10.1093/jac/dkaa267).
We better clarify this issue in Methods section (refer to Line 304-307) and we added specific references (no. 12, 42, and 43).
Round 3
Reviewer 4 Report
Thanks for the resolution of queries to the satisfaction.
minor editings required